# Learned in Translation: Contextualized Word Vectors

**Bryan McCann**
bmccann@salesforce.com

**James Bradbury**
james.bradbury@salesforce.com

**Caiming Xiong**
cxiong@salesforce.com

**Richard Socher**
rsocher@salesforce.com

## Abstract

Computer vision has benefited from initializing multiple deep layers with weights pretrained on large supervised training sets like ImageNet. Natural language processing (NLP) typically sees initialization of only the lowest layer of deep models with pretrained word vectors. In this paper, we use a deep LSTM encoder from an attentional sequence-to-sequence model trained for machine translation (MT) to contextualize word vectors. We show that adding these context vectors (CoVe) improves performance over using only unsupervised word and character vectors on a wide variety of common NLP tasks: sentiment analysis (SST, IMDb), question classification (TREC), entailment (SNLI), and question answering (SQuAD). For fine-grained sentiment analysis and entailment, CoVe improves performance of our baseline models to the state of the art.

## 1 Introduction

Significant gains have been made through transfer and multi-task learning between synergistic tasks. In many cases, these synergies can be exploited by architectures that rely on similar components. In computer vision, convolutional neural networks (CNNs) pretrained on ImageNet [Krizhevsky et al., 2012, Deng et al., 2009] have become the *de facto* initialization for more complex and deeper models. This initialization improves accuracy on other related tasks such as visual question answering [Xiong et al., 2016] or image captioning [Lu et al., 2016, Socher et al., 2014].

In NLP, distributed representations pretrained with models like Word2Vec [Mikolov et al., 2013] and GloVe [Pennington et al., 2014] have become common initializations for the word vectors of deep learning models. Transferring information from large amounts of unlabeled training data in the form of word vectors has shown to improve performance over random word vector initialization on a variety of downstream tasks, e.g. part-of-speech tagging [Collobert et al., 2011], named entity recognition [Pennington et al., 2014], and question answering [Xiong et al., 2017]; however, words rarely appear in isolation. The ability to share a common representation of words in the context of sentences that include them could further improve transfer learning in NLP.

Inspired by the successful transfer of CNNs trained on ImageNet to other tasks in computer vision, we focus on training an encoder for a large NLP task and transferring that encoder to other tasks in NLP. Machine translation (MT) requires a model to encode words in context so as to decode them into another language, and attentional sequence-to-sequence models for MT often contain an LSTM-based encoder, which is a common component in other NLP models. We hypothesize that MT data in general holds potential comparable to that of ImageNet as a cornerstone for reusable models. This makes an MT-LSTM pairing in NLP a natural candidate for mirroring the ImageNet-CNN pairing of computer vision.

As depicted in Figure 1, we begin by training LSTM encoders on several machine translation datasets, and we show that these encoders can be used to improve performance of models trained for other

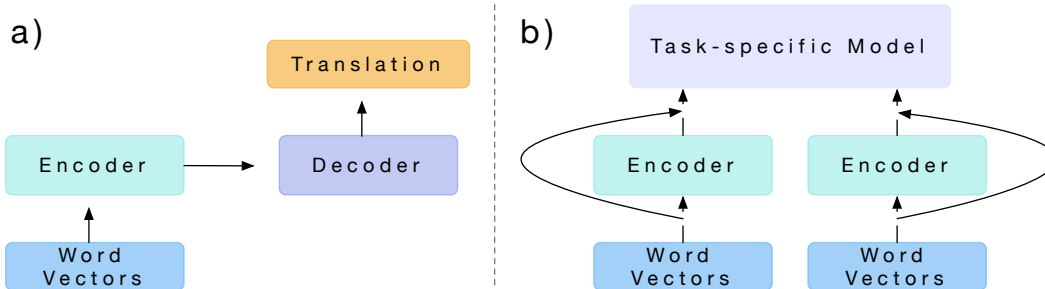

Figure 1: We a) train a two-layer, bidirectional LSTM as the encoder of an attentional sequence-to-sequence model for machine translation and b) use it to provide context for other NLP models.

tasks in NLP. In order to test the transferability of these encoders, we develop a common architecture for a variety of classification tasks, and we modify the Dynamic Coattention Network for question answering [Xiong et al., 2017]. We append the outputs of the MT-LSTMs, which we call context vectors (CoVe), to the word vectors typically used as inputs to these models. This approach improved the performance of models for downstream tasks over that of baseline models using pretrained word vectors alone. For the Stanford Sentiment Treebank (SST) and the Stanford Natural Language Inference Corpus (SNLI), CoVe pushes performance of our baseline model to the state of the art.

Experiments reveal that the quantity of training data used to train the MT-LSTM is positively correlated with performance on downstream tasks. This is yet another advantage of relying on MT, as data for MT is more abundant than for most other supervised NLP tasks, and it suggests that higher quality MT-LSTMs carry over more useful information. This reinforces the idea that machine translation is a good candidate task for further research into models that possess a stronger sense of natural language understanding.

## 2 Related Work

**Transfer Learning.** Transfer learning, or domain adaptation, has been applied in a variety of areas where researchers identified synergistic relationships between independently collected datasets. Saenko et al. [2010] adapt object recognition models developed for one visual domain to new imaging conditions by learning a transformation that minimizes domain-induced changes in the feature distribution. Zhu et al. [2011] use matrix factorization to incorporate textual information into tagged images to enhance image classification. In natural language processing (NLP), Collobert et al. [2011] leverage representations learned from unsupervised learning to improve performance on supervised tasks like named entity recognition, part-of-speech tagging, and chunking. Recent work in NLP has continued in this direction by using pretrained word representations to improve models for entailment [Bowman et al., 2014], sentiment analysis [Socher et al., 2013], summarization [Nallapati et al., 2016], and question answering [Seo et al., 2017, Xiong et al., 2017]. Ramachandran et al. [2016] propose initializing sequence-to-sequence models with pretrained language models and fine-tuning for a specific task. Kiros et al. [2015] propose an unsupervised method for training an encoder that outputs sentence vectors that are predictive of surrounding sentences. We also propose a method of transferring higher-level representations than word vectors, but we use a supervised method to train our sentence encoder and show that it improves models for text classification and question answering without fine-tuning.

**Neural Machine Translation.** Our source domain of transfer learning is machine translation, a task that has seen marked improvements in recent years with the advance of neural machine translation (NMT) models. Sutskever et al. [2014] investigate sequence-to-sequence models that consist of a neural network encoder and decoder for machine translation. Bahdanau et al. [2015] propose the augmenting sequence to sequence models with an attention mechanism that gives the decoder access to the encoder representations of the input sequence at each step of sequence generation. Luong et al. [2015] further study the effectiveness of various attention mechanisms with respect to machine translation. Attention mechanisms have also been successfully applied to NLP tasks like entailment [Conneau et al., 2017], summarization [Nallapati et al., 2016], question answering [Seo et al., 2017, Xiong et al., 2017, Min et al., 2017], and semantic parsing [Dong and Lapata, 2016]. We show that attentional encoders trained for NMT transfer well to other NLP tasks.

**Transfer Learning and Machine Translation.** Machine translation is a suitable source domain for transfer learning because the task, by nature, requires the model to faithfully reproduce a sentence in the target language without losing information in the source language sentence. Moreover, there is an abundance of machine translation data that can be used for transfer learning. Hill et al. [2016] study the effect of transferring from a variety of source domains to the semantic similarity tasks in Agirre et al. [2014]. Hill et al. [2017] further demonstrate that fixed-length representations obtained from NMT encoders outperform those obtained from monolingual (e.g. language modeling) encoders on semantic similarity tasks. Unlike previous work, we do not transfer from fixed length representations produced by NMT encoders. Instead, we transfer representations for each token in the input sequence. Our approach makes the transfer of the trained encoder more directly compatible with subsequent LSTMs, attention mechanisms, and, in general, layers that expect input sequences. This additionally facilitates the transfer of sequential dependencies between encoder states.

**Transfer Learning in Computer Vision.** Since the success of CNNs on the ImageNet challenge, a number of approaches to computer vision tasks have relied on pretrained CNNs as off-the-shelf feature extractors. Girshick et al. [2014] show that using a pretrained CNN to extract features from region proposals improves object detection and semantic segmentation models. Qi et al. [2016] propose a CNN-based object tracking framework, which uses hierarchical features from a pretrained CNN (VGG-19 by Simonyan and Zisserman [2014]). For image captioning, Lu et al. [2016] train a visual sentinel with a pretrained CNN and fine-tune the model with a smaller learning rate. For VQA, Fukui et al. [2016] propose to combine text representations with visual representations extracted by a pretrained residual network [He et al., 2016]. Although model transfer has seen widespread success in computer vision, transfer learning beyond pretrained word vectors is far less pervasive in NLP.

## 3   Machine Translation Model

We begin by training an attentional sequence-to-sequence model for English-to-German translation based on Klein et al. [2017] with the goal of transferring the encoder to other tasks.

For training, we are given a sequence of words in the source language $w^x = [w_1^x, \ldots, w_n^x]$ and a sequence of words in the target language $w^z = [w_1^z, \ldots, w_m^z]$. Let GloVe($w^x$) be a sequence of GloVe vectors corresponding to the words in $w^x$, and let $z$ be a sequence of randomly initialized word vectors corresponding to the words in $w^z$.

We feed GloVe($w^x$) to a standard, two-layer, bidirectional, long short-term memory network [1] [Graves and Schmidhuber, 2005] that we refer to as an MT-LSTM to indicate that it is this same two-layer BiLSTM that we later transfer as a pretrained encoder. The MT-LSTM is used to compute a sequence of hidden states

$$h = \text{MT-LSTM}(\text{GloVe}(w^x)). \tag{1}$$

For machine translation, the MT-LSTM supplies the context for an attentional decoder that produces a distribution over output words $p(\hat{w}_t^z | H, w_1^z, \ldots, w_{t-1}^z)$ at each time-step.

At time-step $t$, the decoder first uses a two-layer, unidirectional LSTM to produce a hidden state $h_t^{\text{dec}}$ based on the previous target embedding $z_{t-1}$ and a context-adjusted hidden state $\tilde{h}_{t-1}$:

$$h_t^{\text{dec}} = \text{LSTM}\left([z_{t-1}; \tilde{h}_{t-1}], h_{t-1}^{\text{dec}}\right). \tag{2}$$

The decoder then computes a vector of attention weights $\alpha$ representing the relevance of each encoding time-step to the current decoder state.

$$\alpha_t = \text{softmax}\left(H(W_1 h_t^{\text{dec}} + b_1)\right) \tag{3}$$

where $H$ refers to the elements of $h$ stacked along the time dimension.

The decoder then uses these weights as coefficients in an attentional sum that is concatenated with the decoder state and passed through a tanh layer to form the context-adjusted hidden state $\tilde{h}$:

$$\tilde{h}_t = \left[ \tanh\left( W_2 H^\top \alpha_t + b_2; h_t^{\text{dec}} \right) \right] \tag{4}$$

The distribution over output words is generated by a final transformation of the context-adjusted hidden state: $p(\hat{w}_t^z | X, w_1^z, \ldots, w_{t-1}^z) = \text{softmax}\left( W_{\text{out}} \tilde{h}_t + b_{\text{out}} \right)$.

## 4 Context Vectors (CoVe)

We transfer what is learned by the MT-LSTM to downstream tasks by treating the outputs of the MT-LSTM as context vectors. If $w$ is a sequence of words and GloVe($w$) the corresponding sequence of word vectors produced by the GloVe model, then

$$\text{CoVe}(w) = \text{MT-LSTM}(\text{GloVe}(w)) \tag{5}$$

is the sequence of context vectors produced by the MT-LSTM. For classification and question answering, for an input sequence $w$, we concatenate each vector in GloVe($w$) with its corresponding vector in CoVe($w$)

$$\tilde{w} = [\text{GloVe}(w); \text{CoVe}(w)] \tag{6}$$

as depicted in Figure 1b.

## 5 Classification with CoVe

We now describe a general biattentive classification network (BCN) we use to test how well CoVe transfer to other tasks. This model, shown in Figure 2, is designed to handle both single-sentence and two-sentence classification tasks. In the case of single-sentence tasks, the input sequence is duplicated to form two sequences, so we will assume two input sequences for the rest of this section.

Input sequences $w^x$ and $w^y$ are converted to sequences of vectors, $\tilde{w}^x$ and $\tilde{w}^y$, as described in Eq. 6 before being fed to the task-specific portion of the model (Figure 1b).

A function $f$ applies a feedforward network with ReLU activation [Nair and Hinton, 2010] to each element of $\tilde{w}^x$ and $\tilde{w}^y$, and a bidirectional LSTM processes the resulting sequences to obtain task specific representations,

$$x = \text{biLSTM}\left( f(\tilde{w}^x) \right) \tag{7}$$

$$y = \text{biLSTM}\left( f(\tilde{w}^y) \right) \tag{8}$$

These sequences are each stacked along the time axis to get matrices $X$ and $Y$.

In order to compute representations that are interdependent, we use a biattention mechanism [Seo et al., 2017, Xiong et al., 2017]. The biattention first computes an affinity matrix $A = XY^\top$. It then extracts attention weights with column-wise normalization:

$$A_x = \text{softmax}\left( A \right) \qquad A_y = \text{softmax}\left( A^\top \right) \tag{9}$$

which amounts to a novel form of self-attention when $x = y$. Next, it uses context summaries

$$C_x = A_x^\top X \qquad C_y = A_y^\top Y \tag{10}$$

to condition each sequence on the other.

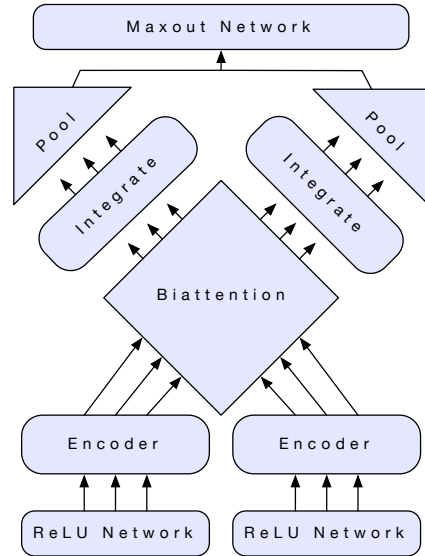

Figure 2: Our BCN uses a feedforward network with ReLU activation and biLSTM encoder to create task-specific representations of each input sequence. Biattention conditions each representation on the other, a biLSTM integrates the conditional information, and a maxout network uses pooled features to compute a distribution over possible classes.

We integrate the conditioning information into our representations for each sequence with two separate one-layer, bidirectional LSTMs that operate on the concatenation of the original representations (to ensure no information is lost in conditioning), their differences from the context summaries (to explicitly capture the difference from the original signals), and the element-wise products between originals and context summaries (to amplify or dampen the original signals).

$$X_{|y} = \text{biLSTM}\left([X; X - C_y; X \odot C_y]\right) \tag{11}$$
$$Y_{|x} = \text{biLSTM}\left([Y; Y - C_x; Y \odot C_x]\right) \tag{12}$$

The outputs of the bidirectional LSTMs are aggregated by pooling along the time dimension. Max and mean pooling have been used in other models to extract features, but we have found that adding both min pooling and self-attentive pooling can aid in some tasks. Each captures a different perspective on the conditioned sequences.

The self-attentive pooling computes weights for each time step of the sequence

$$\beta_x = \text{softmax}\left(X_{|y} v_1 + d_1\right) \qquad \beta_y = \text{softmax}\left(Y_{|x} v_2 + d_2\right) \tag{13}$$

and uses these weights to get weighted summations of each sequence:

$$x_{\text{self}} = X_{|y}^\top \beta_x \qquad y_{\text{self}} = Y_{|x}^\top \beta_y \tag{14}$$

The pooled representations are combined to get one joined representation for all inputs.

$$x_{\text{pool}} = \left[\max(X_{|y}); \text{mean}(X_{|y}); \min(X_{|y}); x_{\text{self}}\right] \tag{15}$$
$$y_{\text{pool}} = \left[\max(Y_{|x}); \text{mean}(Y_{|x}); \min(Y_{|x}); y_{\text{self}}\right] \tag{16}$$

We feed this joined representation through a three-layer, batch-normalized [Ioffe and Szegedy, 2015] maxout network [Goodfellow et al., 2013] to produce a probability distribution over possible classes.

# 6 Question Answering with CoVe

For question answering, we obtain sequences $x$ and $y$ just as we do in Eq. 7 and Eq. 8 for classification, except that the function $f$ is replaced with a function $g$ that uses a tanh activation instead of a ReLU activation. In this case, one of the sequences is the document and the other the question in the question-document pair. These sequences are then fed through the coattention and dynamic decoder implemented as in the original Dynamic Coattention Network (DCN) [Xiong et al., 2016].

# 7 Datasets

**Machine Translation.** We use three different English-German machine translation datasets to train three separate MT-LSTMs. Each is tokenized using the Moses Toolkit [Koehn et al., 2007].

Our smallest MT dataset comes from the WMT 2016 multi-modal translation shared task [Specia et al., 2016]. The training set consists of 30,000 sentence pairs that briefly describe Flickr captions and is often referred to as Multi30k. Due to the nature of image captions, this dataset contains sentences that are, on average, shorter and simpler than those from larger counterparts.

Our medium-sized MT dataset is the 2016 version of the machine translation task prepared for the International Workshop on Spoken Language Translation [Cettolo et al., 2015]. The training set consists of 209,772 sentence pairs from transcribed TED presentations that cover a wide variety of topics with more conversational language than in the other two machine translation datasets.

Our largest MT dataset comes from the news translation shared task from WMT 2017. The training set consists of roughly 7 million sentence pairs that comes from web crawl data, a news and commentary corpus, European Parliament proceedings, and European Union press releases.

We refer to the three MT datasets as MT-Small, MT-Medium, and MT-Large, respectively, and we refer to context vectors from encoders trained on each in turn as CoVe-S, CoVe-M, and CoVe-L.

| Dataset | Task | Details | Examples |
|---------|------|---------|----------|
| SST-2 | Sentiment Classification | 2 classes, single sentences | 56.4k |
| SST-5 | Sentiment Classification | 5 classes, single sentences | 94.2k |
| IMDb | Sentiment Classification | 2 classes, multiple sentences | 22.5k |
| TREC-6 | Question Classification | 6 classes | 5k |
| TREC-50 | Question Classification | 50 classes | 5k |
| SNLI | Entailment Classification | 2 classes | 550k |
| SQuAD | Question Answering | open-ended (answer-spans) | 87.6k |

Table 1: Datasets, tasks, details, and number of training examples.

**Sentiment Analysis.** We train our model separately on two sentiment analysis datasets: the Stanford Sentiment Treebank (SST) [Socher et al., 2013] and the IMDb dataset [Maas et al., 2011]. Both of these datasets comprise movie reviews and their sentiment. We use the binary version of each dataset as well as the five-class version of SST. For training on SST, we use all sub-trees with length greater than 3. SST-2 contains roughly $56,400$ reviews after removing "neutral" examples. SST-5 contains roughly $94,200$ reviews and does include "neutral" examples. IMDb contains $25,000$ multi-sentence reviews, which we truncate to the first 200 words. $2,500$ reviews are held out for validation.

**Question Classification.** For question classification, we use the small TREC dataset [Voorhees and Tice, 1999] dataset of open-domain, fact-based questions divided into broad semantic categories. We experiment with both the six-class and fifty-class versions of TREC, which which refer to as TREC-6 and TREC-50, respectively. We hold out $452$ examples for validation and leave $5,000$ for training.

**Entailment.** For entailment, we use the Stanford Natural Language Inference Corpus (SNLI) [Bowman et al., 2015], which has 550,152 training, 10,000 validation, and 10,000 testing examples. Each example consists of a premise, a hypothesis, and a label specifying whether the premise entails, contradicts, or is neutral with respect to the hypothesis.

**Question Answering.** The Stanford Question Answering Dataset (SQuAD) [Rajpurkar et al., 2016] is a large-scale question answering dataset with 87,599 training examples, 10,570 development examples, and a test set that is not released to the public. Examples consist of question-answer pairs associated with a paragraph from the English Wikipedia. SQuAD examples assume that the question is answerable and that the answer is contained verbatim somewhere in the paragraph.

# 8 Experiments

## 8.1 Machine Translation

The MT-LSTM trained on MT-Small obtains an uncased, tokenized BLEU score of $38.5$ on the Multi30k test set from 2016. The model trained on MT-Medium obtains an uncased, tokenized BLEU score of $25.54$ on the IWSLT test set from 2014. The MT-LSTM trained on MT-Large obtains an uncased, tokenized BLEU score of $28.96$ on the WMT 2016 test set. These results represent strong baseline machine translation models for their respective datasets. Note that, while the smallest dataset has the highest BLEU score, it is also a much simpler dataset with a restricted domain.

**Training Details.** When training an MT-LSTM, we used fixed 300-dimensional word vectors. We used the CommonCrawl-840B GloVe model for English word vectors, which were completely fixed during training, so that the MT-LSTM had to learn how to use the pretrained vectors for translation. The hidden size of the LSTMs in all MT-LSTMs is 300. Because all MT-LSTMs are bidirectional, they output 600-dimensional vectors. The model was trained with stochastic gradient descent with a learning rate that began at 1 and decayed by half each epoch after the validation perplexity increased for the first time. Dropout with ratio $0.2$ was applied to the inputs and outputs of all layers of the encoder and decoder.

## 8.2 Classification and Question Answering

For classification and question answering, we explore how varying the input representations affects final performance. Table 2 contains validation performances for experiments comparing the use of GloVe, character n-grams, CoVe, and combinations of the three.

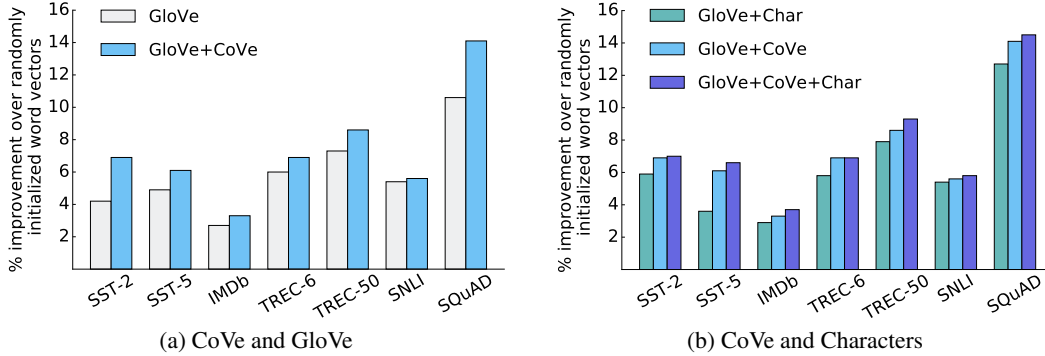

(a) CoVe and GloVe          (b) CoVe and Characters

Figure 3: The Benefits of CoVe

| Dataset | Random | GloVe | GloVe+ | | | | |
|---|---|---|---|---|---|---|---|
| | | | Char | CoVe-S | CoVe-M | CoVe-L | Char+CoVe-L |
| SST-2 | 84.2 | 88.4 | 90.1 | 89.0 | 90.9 | 91.1 | **91.2** |
| SST-5 | 48.6 | 53.5 | 52.2 | 54.0 | 54.7 | 54.5 | **55.2** |
| IMDb | 88.4 | 91.1 | 91.3 | 90.6 | 91.6 | 91.7 | **92.1** |
| TREC-6 | 88.9 | 94.9 | 94.7 | 94.7 | 95.1 | 95.8 | **95.8** |
| TREC-50 | 81.9 | 89.2 | 89.8 | 89.6 | 89.6 | 90.5 | **91.2** |
| SNLI | 82.3 | 87.7 | 87.7 | 87.3 | 87.5 | 87.9 | **88.1** |
| SQuAD | 65.4 | 76.0 | 78.1 | 76.5 | 77.1 | 79.5 | **79.9** |

Table 2: CoVe improves validation performance. CoVe has an advantage over character n-gram embeddings, but using both improves performance further. Models benefit most by using an MT-LSTM trained with MT-Large (CoVe-L). Accuracy is reported for classification tasks, and F1 is reported for SQuAD.

**Training Details.** Unsupervised vectors and MT-LSTMs remain fixed in this set of experiments. LSTMs have hidden size 300. Models were trained using Adam with $\alpha = 0.001$. Dropout was applied before all feedforward layers with dropout ratio $0.1$, $0.2$, or $0.3$. Maxout networks pool over $4$ channels, reduce dimensionality by $2$, $4$, or $8$, reduce again by $2$, and project to the output dimension.

**The Benefits of CoVe.** Figure 3a shows that models that use CoVe alongside GloVe achieve higher validation performance than models that use only GloVe. Figure 3b shows that using CoVe in Eq. 6 brings larger improvements than using character n-gram embeddings [Hashimoto et al., 2016]. It also shows that altering Eq. 6 by additionally appending character n-gram embeddings can boost performance even further for some tasks. This suggests that the information provided by CoVe is complementary to both the word-level information provided by GloVe as well as the character-level information provided by character n-gram embeddings.

**The Effects of MT Training Data.** We experimented with different training datasets for the MT-LSTMs to see how varying the MT training data affects the benefits of using CoVe in downstream tasks. Figure 4 shows an important trend we can extract from Table 2. There appears to be a positive correlation between the larger MT datasets, which contain more complex, varied language, and the improvement that using CoVe brings to downstream tasks. This is evidence for our hypothesis that MT data has potential as a large resource for transfer learning in NLP.

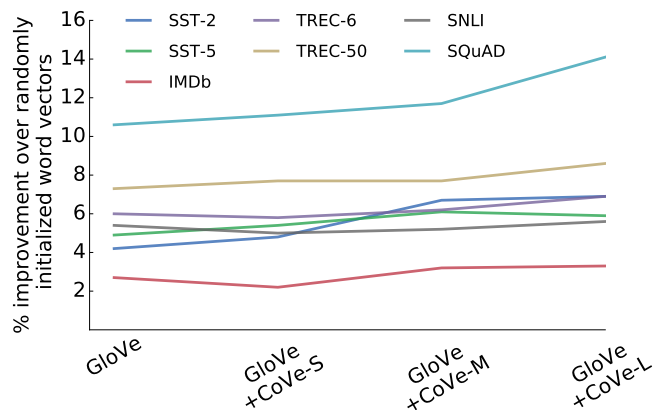

Figure 4: The Effects of MT Training Data

| | Model | Test | | Model | Test |
|---|---|---|---|---|---|
| SST-2 | P-LSTM [Wieting et al., 2016] | 89.2 | TREC-6 | SVM [da Silva et al., 2011] | 95.0 |
| | CT-LSTM [Looks et al., 2017] | 89.4 | | SVM [Van-Tu and Anh-Cuong, 2016] | 95.2 |
| | TE-LSTM [Huang et al., 2017] | 89.6 | | DSCNN-P [Zhang et al., 2016] | 95.6 |
| | NSE [Munkhdalai and Yu, 2016a] | 89.7 | | *BCN+Char+CoVe [Ours]* | *95.8* |
| | *BCN+Char+CoVe [Ours]* | *90.3* | | TBCNN [Mou et al., 2015] | 96.0 |
| | **bmLSTM [Radford et al., 2017]** | **91.8** | | **LSTM-CNN [Zhou et al., 2016]** | **96.1** |
| SST-5 | MVN [Guo et al., 2017] | 51.5 | TREC-50 | SVM [Loni et al., 2011] | 89.0 |
| | DMN [Kumar et al., 2016] | 52.1 | | SNoW [Li and Roth, 2006] | 89.3 |
| | LSTM-CNN [Zhou et al., 2016] | 52.4 | | *BCN+Char+CoVe [Ours]* | *90.2* |
| | TE-LSTM [Huang et al., 2017] | 52.6 | | RulesUHC [da Silva et al., 2011] | 90.8 |
| | NTI [Munkhdalai and Yu, 2016b] | 53.1 | | SVM [Van-Tu and Anh-Cuong, 2016] | 91.6 |
| | *BCN+Char+CoVe [Ours]* | *53.7* | | **Rules [Madabushi and Lee, 2016]** | **97.2** |
| IMDb | *BCN+Char+CoVe [Ours]* | *91.8* | SNLI | DecAtt+Intra [Parikh et al., 2016] | 86.8 |
| | SA-LSTM [Dai and Le, 2015] | 92.8 | | NTI [Munkhdalai and Yu, 2016b] | 87.3 |
| | bmLSTM [Radford et al., 2017] | 92.9 | | re-read LSTM [Sha et al., 2016] | 87.5 |
| | TRNN [Dieng et al., 2016] | 93.8 | | btree-LSTM [Paria et al., 2016] | 87.6 |
| | oh-LSTM [Johnson and Zhang, 2016] | 94.1 | | 600D ESIM [Chen et al., 2016] | 88.0 |
| | **Virtual [Miyato et al., 2017]** | **94.1** | | *BCN+Char+CoVe [Ours]* | *88.1* |

Table 4: Single model test accuracies for classification tasks.

**Test Performance.** Table 4 shows the final test accuracies of our best classification models, each of which achieved the highest validation accuracy on its task using GloVe, CoVe, and character n-gram embeddings. Final test performances on SST-5 and SNLI reached a new state of the art.

Table 3 shows how the validation exact match and F1 scores of our best SQuAD model compare to the scores of the most recent top models in the literature. We did not submit the SQuAD model for testing, but the addition of CoVe was enough to push the validation performance of the original DCN, which already used character n-gram embeddings, above the validation performance of the published version of the R-NET. Test performances are tracked by the SQuAD leaderboard [2].

| Model | EM | F1 |
|---|---|---|
| LR [Rajpurkar et al., 2016] | 40.0 | 51.0 |
| DCR [Yu et al., 2017] | 62.5 | 72.1 |
| hM-LSTM+AP [Wang and Jiang, 2017] | 64.1 | 73.9 |
| DCN+Char [Xiong et al., 2017] | 65.4 | 75.6 |
| BiDAF [Seo et al., 2017] | 68.0 | 77.3 |
| R-NET [Wang et al., 2017] | 71.1 | 79.5 |
| *DCN+Char+CoVe [Ours]* | *71.3* | *79.9* |

Table 3: Exact match and F1 validation scores for single-model question answering.

**Comparison to Skip-Thought Vectors.** Kiros et al. [2015] show how to encode a sentence into a single skip-thought vector that transfers well to a variety of tasks. Both skip-thought and CoVe pretrain encoders to capture information at a higher level than words. However, skip-thought encoders are trained with an unsupervised method that relies on the final output of the encoder. MT-LSTMs are trained with a supervised method that instead relies on intermediate outputs associated with each input word. Additionally, the 4800 dimensional skip-thought vectors make training more unstable than using the 600 dimensional CoVe. Table 5 shows that these differences make CoVe more suitable for transfer learning in our classification experiments.

| | GloVe+Char+ | |
|---|---|---|
| Dataset | Skip-Thought | CoVe-L |
| SST-2 | 88.7 | **91.2** |
| SST-5 | 52.1 | **55.2** |
| TREC-6 | 94.2 | **95.8** |
| TREC-50 | 89.6 | **91.2** |
| SNLI | 86.0 | **88.1** |

Table 5: Classification validation accuracies with skip-thought and CoVe.

---

# 9 Conclusion

We introduce an approach for transferring knowledge from an encoder pretrained on machine translation to a variety of downstream NLP tasks. In all cases, models that used CoVe from our best, pretrained MT-LSTM performed better than baselines that used random word vector initialization, baselines that used pretrained word vectors from a GloVe model, and baselines that used word vectors from a GloVe model together with character n-gram embeddings. We hope this is a step towards the goal of building unified NLP models that rely on increasingly more general reusable weights.

The PyTorch code at `https://github.com/salesforce/cove` includes an example of how to generate CoVe from the MT-LSTM we used in all of our best models. We hope that making our best MT-LSTM available will encourage further research into shared representations for NLP models.

## Footnotes

[1] Since there are several biLSTM variants, we define ours as follows. Let $h = [h_1, \ldots, h_n] = \text{biLSTM}(x)$ represent the output sequence of our biLSTM operating on an input sequence $x$. Then a forward LSTM computes $\overrightarrow{h_t} = \text{LSTM}(x_t, \overrightarrow{h_{t-1}})$ for each time step, and a backward LSTM computes $\overleftarrow{h_t} = \text{LSTM}(x_t, \overleftarrow{h_{t+1}})$. The final outputs of the biLSTM for each time step are $h_t = [\overrightarrow{h_t}; \overleftarrow{h_t}]$.

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
