[Reviews · NeurIPS 2017]

Reviewer 1



This paper proposes to pretrain sentence encoders for various NLP tasks using machine translation data. In particular, the authors propose to share the whole pretrained sentence encoding model (an LSTM with attention), not just the word embeddings as have been done to great success over the last few years. Evaluations are carried out on sentence classification tasks (sentiment, entailment & question classification) as well as question answering. In all evaluations, the pretrained model outperforms a randomly initialized model with pretrained GloVe embeddings only. This is a good paper that presents a simple idea in a clear, easy to understand manner. The fact that the pretraining helps all tasks across the board is an important finding, and therefore I recommend this paper for publication. Are the authors going to release their pretrained models? The authors should cite https://arxiv.org/pdf/1611.02683.pdf, where the encoders are pretrained as language models.

Reviewer 2



This paper explores the use of attention based sequence to sequence models for machine translation to pretrain models for other tasks. The author evaluated their methods on a wide range of tasks (entailment, sentiment, question classification and question answering). Their results showed consistent improvements over the baselines. Overall, this paper is well written and the idea is interesting. In my opinion, the contributions of this paper are expected (moving from pretrained word embeddings to pretrained sentence embeddings) but important for sentence classification. I have the following suggestions to improve the paper: - The authors should have a better explaination why the performance could not be improved for the entailment task while increasing the amount of MT training pairs. - From my experiences, I always expected to see improvements using character n-gram embeddings. Why is that not the case in this paper?

Reviewer 3



The paper proposes uses to reuse the weights of an LSTM that was trained as a part of a neural machine translator to initialize LSTMs trained for downstream tasks. The approach brings improvement on classification, semantic entailment recognition and question answering tasks. The paper is clearly written and easy to understand. The evaluation is exhaustive and convincing. The paper is solid and should be accepted. A few suggestion for improving the paper even further: - I think it would be better to use smaller subsets of the largest MT dataset available instead of using three different datasets. This way the influence of domain mismatch and the amount of data available would be disentangled. - it would be great to see a comparison with a skip-thought-trained LSTM and with an LSTM that was pretrained as a language model. - I encourage the authors to release their evaluation code to fascilitate the research on trasfer learning for NLP.